# Soluble P-tau217 reflects amyloid and tau pathology and mediates the association of amyloid with tau

Niklas Mattsson-Carlgren[1,2,3,*] (iD), Shorena Janelidze[1], Randall J Bateman[4], Ruben Smith[1], Erik Stomrud[1,5], Geidy E Serrano[6], Eric M Reiman[6], Sebastian Palmqvist[1,5] (iD), Jeffrey L Dage[7], Thomas G Beach[8] & Oskar Hansson[1,4,5,**] (iD)

## Abstract

Alzheimer's disease is characterized by β-amyloid plaques and tau tangles. Plasma levels of phospho-tau217 (P-tau217) accurately differentiate Alzheimer's disease dementia from other dementias, but it is unclear to what degree this reflects β-amyloid plaque accumulation, tau tangle accumulation, or both. In a cohort with post-mortem neuropathological data ($N = 88$), both plaque and tangle density contributed independently to higher P-tau217, but P-tau217 was not elevated in patients with non-Alzheimer's disease tauopathies ($N = 9$). Several findings were replicated in a cohort with PET imaging ("BioFINDER-2", $N = 426$), where β-amyloid and tau PET were independently associated with P-tau217. P-tau217 concentrations correlated with β-amyloid PET (but not tau PET) in early disease stages and with both β-amyloid and (more strongly) tau PET in late disease stages. Finally, P-tau217 mediated the association between β-amyloid and tau in both cohorts, especially for tau outside of the medial temporal lobe. These findings support the hypothesis that plasma P-tau217 concentration is increased by both β-amyloid plaques and tau tangles and is congruent with the hypothesis that P-tau is involved in β-amyloid-dependent formation of neocortical tau tangles.

**Keywords** Alzheimer's disease; amyloid; phosphorylated tau; plasma; tau
**Subject Categories** Biomarkers; Neuroscience

## Introduction

β-amyloid (Aβ) pathology and tau pathology are core hallmarks of Alzheimer's disease (AD). The amyloid cascade hypothesis states that altered Aβ metabolism is an initiating event, which drives accumulation of aggregated tau (Sperling *et al*, 2014). These events are associated with altered levels of fluid biomarkers, as has been extensively studied in both cerebrospinal fluid (CSF) and plasma (Olsson *et al*, 2016; Hampel *et al*, 2018). Recently, it has been shown that plasma levels of hyperphosphorylated tau (including both P-tau181 (Janelidze *et al*, 2020; Karikari *et al*, 2020) and P-tau217 (Palmqvist *et al*, 2020)) are markedly elevated in patients with AD. Plasma P-tau217 levels have particularly high accuracy for AD and are correlated with tangle densities in post-mortem data (Palmqvist *et al*, 2020). However, recent studies have suggested that CSF P-tau217 (and other P-tau variants) can be increased as a function of Aβ accumulation (Suárez-Calvet *et al*, 2020), also in individuals who are tau-negative as determined by positron emission tomography (PET) (Barthélemy *et al*, 2020b; Mattsson-Carlgren *et al*, 2020) (but note that tau PET has limited sensitivity to detect very sparse tau aggregation (Fleisher *et al*, 2020)). It is unclear to what degree plasma P-tau217 levels depend on Aβ versus tau load in the brain. Here, we set out to clarify this question in two cohorts, including a cohort with post-mortem quantification of Aβ plaques and tau tangles, and a large prospective cohort where Aβ and tau load were measured *in vivo* using PET imaging. In sum, we found converging evidence from both cohorts that Aβ and tau load had independent and interactive effects on plasma P-tau217. We also found that plasma P-tau217 levels statistically mediated the effect of Aβ load on tau load. Finally, we found that in subjects with very limited tangle pathology, Aβ plaque load but not medial temporal lobe tangle load

1  Clinical Memory Research Unit, Faculty of Medicine, Lund University, Lund, Sweden
2  Department of Neurology, Skåne University Hospital, Lund University, Lund, Sweden
3  Wallenberg Center for Molecular Medicine, Lund University, Lund, Sweden
4  Department of Neurology, Washington University School of Medicine, Saint-Louis, MO, USA
5  Memory Clinic, Skåne University Hospital, Malmö, Sweden
6  Banner Alzheimer's Institute, Phoenix, AZ, USA
7  Eli Lilly and Company, Indianapolis, IN, USA
8  Banner Sun Health Research Institute, Sun City, AZ, USA
   *Corresponding author. Tel: +46 072 575 9329; E-mail: niklas.mattsson-carlgren@med.lu.se
   **Corresponding author. Tel: +46 072 226 7745; E-mail: oskar.hansson@med.lu.se

remained statistically associated with plasma P-tau217. Taken together, the data from this observational study support the theory that the first increases in release and phosphorylation of soluble tau measured in plasma are driven by Aβ aggregation and may appear largely before spread of tau tangles outside of the medial temporal lobe. One possibility is that soluble P-tau217 plays a role in the spread of neocortical tangles, which could motivate trials with treatments targeting soluble phosphorylated tau, to break the link between Aβ and tau aggregation in AD.

# Results

### Plasma P-tau217 is independently associated with pathology measures of both Aβ and tau

We first tested associations between antemortem plasma P-tau217 and neuropathological quantification of Aβ plaques and tau tangles in a cohort with post-mortem data ($N = 88$). We compared regression models with plasma P-tau217 as response variable and different sets of predictors, including I) only Aβ plaque density, II) only tau tangle density, III) both Aβ plaque and tangle densities, and IV) Aβ plaque and tangle densities and their interaction. All models included age and sex as covariates. For the individual models, the tangle only model had slightly higher overall explanatory power ($R^2 = 0.47$) than the plaque only model ($R^2 = 0.40$) ($\Delta AIC = 9$ favoring the tangle only model, but the difference in $R^2$

was not significant, tested in a bootstrap procedure, $\Delta R^2 = 0.07$, 95% CI −0.17 to 0.25). Model comparisons favored including both plaques and tangle densities in an interaction model (Fig 1A, $\Delta AIC = 5$-29 compared to the other models). In the interaction model, both plaque density (β = 2.00, $P < 0.0001$; compared to β = 2.95, $P < 0.0001$ in the plaque only model) and tangle density (β = 1.46, $P = 0.0072$; compared to β = 3.29, $P < 0.0001$ in the tangle only model) and the plaque by tangle interaction (β = 1.26, $P = 0.012$) were significantly associated with higher plasma P-tau217 levels. Age ($P = 0.19$) and sex ($P = 0.59$) were non-significant in the interaction model. Higher plaque density had the strongest association with high plasma P-tau217 among those with higher tangle density (visualized with tertiles of tangles in Fig 1B), and higher tangle density had the strongest association with high plasma P-tau217 among those with higher plaque density (visualized with tertiles of plaques in Fig 1C). For all these analyses, plaque and tangle densities were used as continuous variables, centered, and scaled to z-scores. These results supported the hypothesis that plasma P-tau217 is increased by independent effects of both Aβ and tau pathology.

### Plasma P-tau217 statistically mediates the effect of Aβ on tau

We next conducted statistical mediation analyses in the neuropathology cohort (Fig 2). Plasma P-tau217 partly mediated the effect of Aβ plaque density on tau tangle density (but Aβ plaque density retained a significant association with tangle density also in

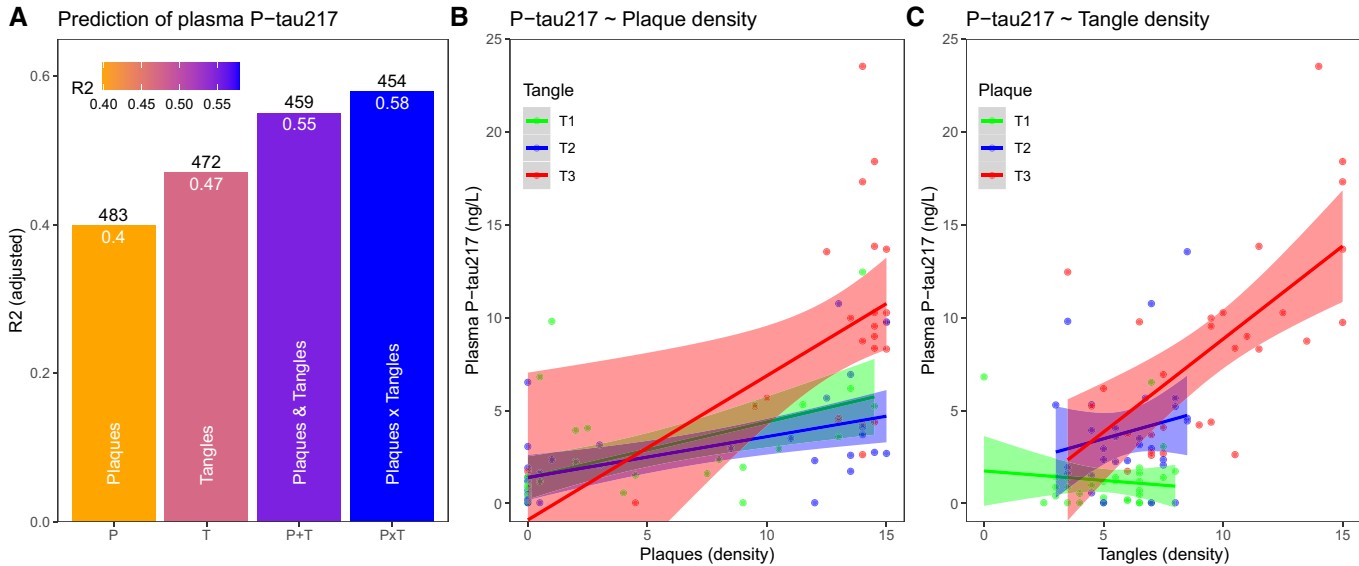

**Figure 1. Independent effects of plaque and tangle density on plasma P-tau217.**

A   $R^2$ and AIC for different regression models, using either only amyloid plaque density ("P"), only tangle density ("T"), both plaque and tangle density ("P + T"), or both plaque and tangle density including their interaction term ("PxT"). All models included age and sex as covariates. The panel shows adjusted $R^2$, together with AIC (above the bars) for each model. We compared $R^2$ between the models using a bootstrap procedure ($N = 1,000$ iterations), which showed that the $R^2$ for the "PxT" model was marginally higher than for the "P + T" model ($\Delta R^2 = 0.027$, 95% CI −0.0034-0.087) and significantly higher than for the "T" model ($\Delta R^2 = 0.12$, 95% CI 0.04-0.24) and the "P" model ($\Delta R^2 = 0.18$, 95% CI 0.055-0.31).

B   Associations between plasma P-tau217 and plaque density (stratified by tertiles [T1-3] of tangle density).

C   Associations between plasma P-tau217 and tangle density (stratified by tertiles [T1-3] of plaque density).

Data information: Tertiles were chosen to visualize the data in panels (B, C), but the regression models used continuous density data as predictors. The solid lines represent mean effects from regression models. The shaded areas are 95% confidence intervals for the mean effects. Samples were analyzed in duplicates.

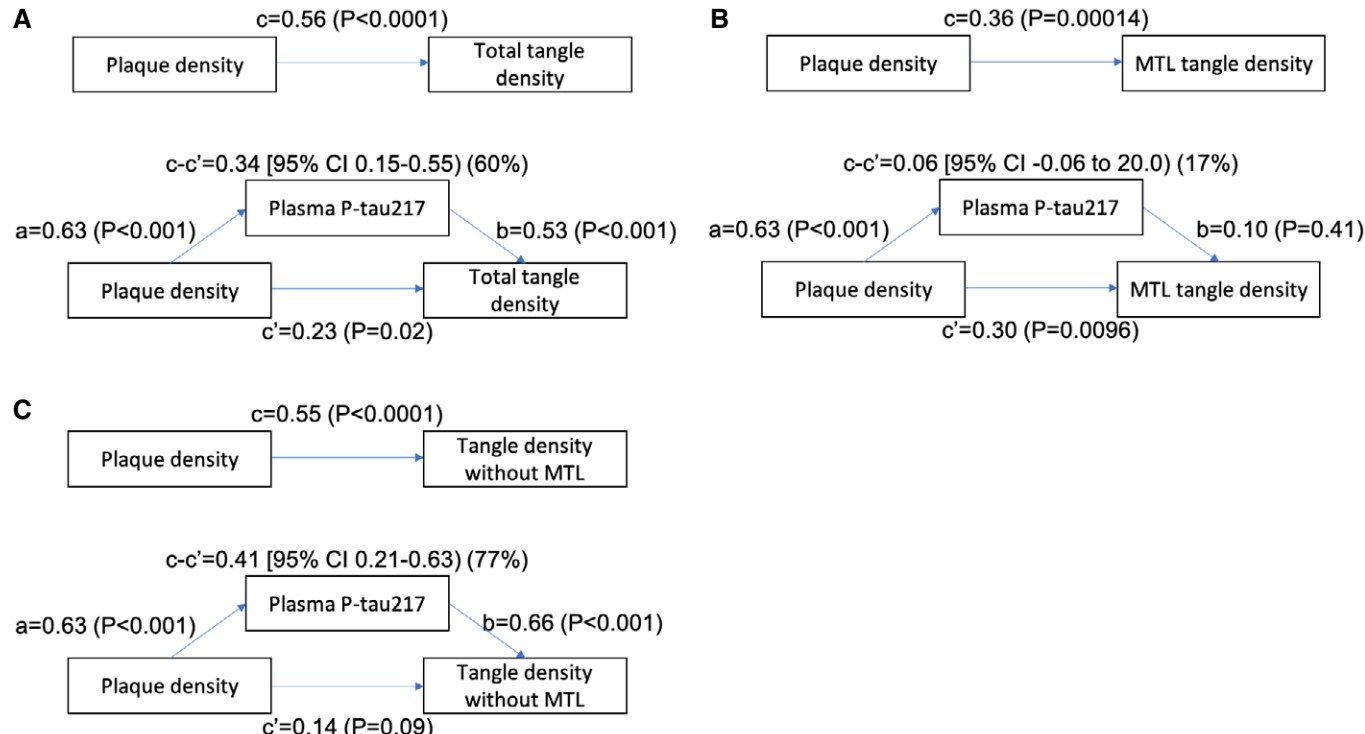

**Figure 2. Plasma P-tau217 mediates effects of plaques on tangles.**

A–C Three different mediation models are shown, testing plasma P-tau217 as a statistical mediator of plaque density on tangle density. Panel (A) shows the mediation of total tangle density (significant mediation, 60% of the direct effect of plaque density on tangle density was explained by plasma P-tau217). Panel (B) shows the mediation of medial temporal lobe (MTL, entorhinal cortex and hippocampus) tangle density (no mediation). Panel (C) shows the mediation of tangle density in all regions except the MTL (strongest mediation, 77%). The *P*-values are from linear regression models used to estimate the mediation effects.

the presence of plasma P-tau217) (60% mediation, Fig 2A). In contrast, there was no significant mediation of effects of Aβ plaque density on tau tangle density in the medial temporal lobe (entorhinal cortex and hippocampus) (the direct effect of Aβ plaques on tau tangles was also weaker, Fig 2B). The strongest mediation was seen for total tangle density when removing tangles in the medial temporal lobe (77% mediation; the direct effect of Aβ plaques on tangles became non-significant, Fig 2C). These results suggest that plasma P-tau217 may partly explain the link between build-up of Aβ and tau pathology, especially for tau pathology that extends over the neocortex, beyond the medial temporal lobe.

**No associations between plasma P-tau217 and isolated medial temporal lobe tau**

To study the effects of very subtle tau pathology on plasma P-tau217, we did a subset analysis on the neuropathology cohort participants without tangles in the parietal or frontal lobe (and only minimal tangle pathology in the temporal lobe, allowing score 0.5, indicating a density halfway between "none" [0] and "sparse" [1]). In this subset ($N = 42$), Aβ plaque density remained significantly associated with plasma P-tau217 ($\beta = 1.64$, $P < 0.0001$) (Fig 3A), but medial temporal lobe tangles were not associated with plasma P-tau217 ($P = 0.40$, Fig 3B) and there was no association between Aβ plaque density and medial temporal lobe tangle density ($P = 0.59$, Fig 3C). These results suggest that Aβ pathology is linked

to increased phosphorylation and/or release of tau even in very early stages of AD (without prominent tangle pathology in the neocortex). In contrast, tau pathology which is contained in the medial temporal lobe without significant neocortical involvement appears to possibly occur independently of Aβ pathology and without altered phosphorylation or release of soluble tau.

**No associations between plasma P-tau217 and primary tauopathies (CBD and PSP)**

To test whether plasma P-tau217 was elevated in individuals with a primary non-AD tauopathy, we performed a sensitivity analysis that included nine additional subjects with neuropathology data and a primary neuropathological diagnosis of PSP ($N = 7$) or CBD ($N = 2$) (7 females/2 males; mean age [SD] at death: 87 [8.7] years; mean time between sample collection and death: 1.1 [0.75] years). These were included from the same neuropathology program as the main pathology cohort. Figure 4 shows plasma P-tau217 (Fig 4A), plaque density (Fig 4B), and tangle density (Fig 4C) for these subjects and (for comparison) for subjects with no significant primary pathology, AD, Parkinson's disease, or other primary pathologies. PSP/CBD subjects had low plasma P-tau217 (no difference compared to subjects without significant primary pathology; $P = 0.66$, Mann–Whitney U-test). Further, they also had low plaque density (no difference compared to subjects without significant primary pathology; $P = 0.13$) and a trend to elevated tangle counts

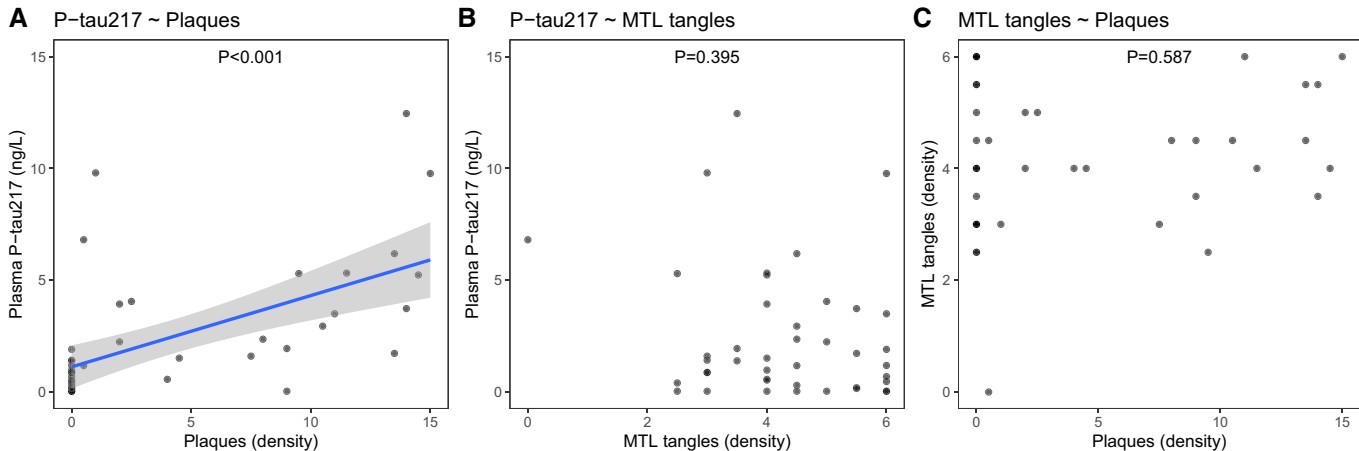

**Figure 3. Associations between plasma P-tau217, Aβ plaque density, and medial temporal lobe (MTL) tangle density in people with limited tau pathology.**

A–C  Individuals in the neuropathology cohort without tangles in the parietal or frontal lobe and no more than minimal tangle load in the temporal lobe were included in this analysis (N = 42). MTL tangles were defined as tangles in entorhinal cortex plus hippocampus. Relationships between variables were tested in linear regression models, adjusted for age and sex. In these models, plasma P-tau217 was significantly associated with Aβ plaques (β = 1.64, P < 0.0001) (panel A) but not with MTL tangles (panel B). MTL tangles were not associated with Aβ plaques (panel C). The solid line in panel (A) is the mean effect from the regression model, and the shaded area is the 95% confidence interval of the mean.

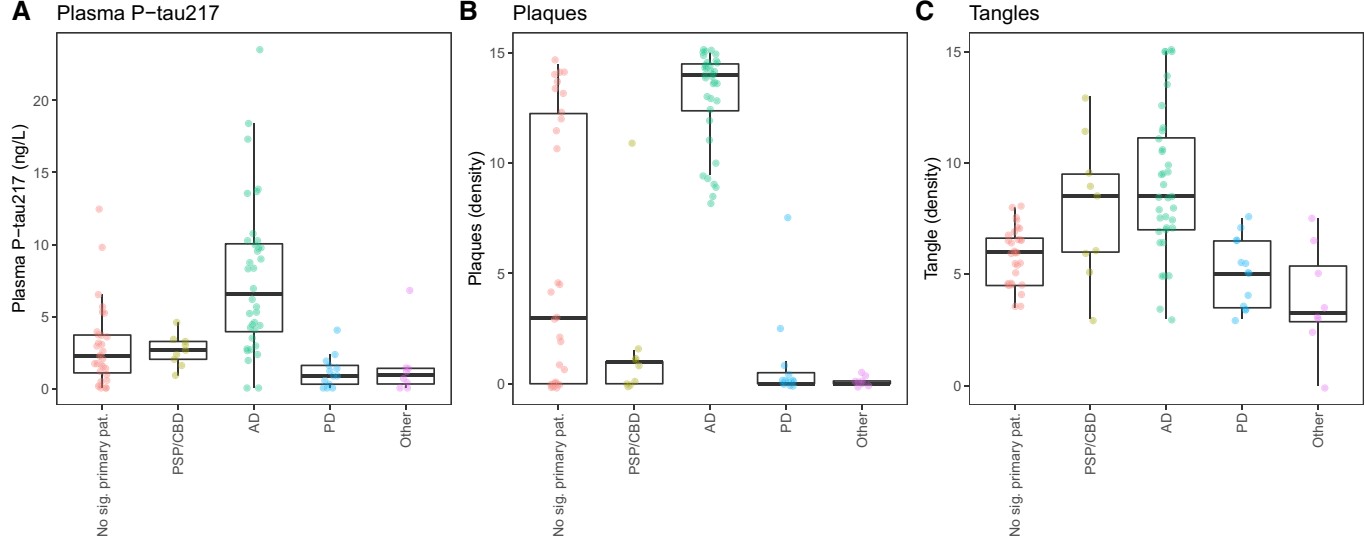

**Figure 4. Plasma P-tau217, Aβ plaque density, and tau tangle density by primary neuropathology.**

A–C  Individuals in the neuropathology cohort were divided into groups of no significant primary pathology (N = 31), AD (N = 36), PD (N = 13), and others (N = 8) and analyzed together with N = 9 additional subjects with primary non-AD tauopathies (PSP/CBD). Data are shown for plasma P-tau217 (panel A), plaque density (panel B), and tangle density (panel C). Plasma P-tau217 was not elevated in PSP/CBD compared to subjects without significant primary pathology. AD, Alzheimer's disease; CBD, corticobasal degeneration; PSP, progressive supranuclear palsy; and PD, Parkinson's disease. The central band of the boxes is medians, and the boxes show interquartile ranges. The whiskers are defined as the smallest (largest) observation greater (less) than or equal to the first (third) quartile minus (plus) 1.5 times the interquartile range.

(higher levels compared to subjects without significant primary pathology, P = 0.068; no difference compared to AD, P = 0.37). This suggests that plasma P-tau217 is not necessarily elevated due to tau accumulation in pure non-AD tauopathies, supporting that the increase in plasma P-tau217 is specific to AD (and may require plaque deposition in addition to tau accumulation).

**Plasma P-tau217 is independently associated with PET measures of both Aβ and tau**

We next tested associations between plasma P-tau and measures of aggregated Aβ and tau in vivo in the BioFINDER-2 cohort (see Table 1). We again compared regression models with plasma

**Table 1. Demographics**

| | Arizona neuropathology cohort | BioFINDER-2 |
|---|---|---|
| N | 88 | 426 |
| Men/female | 35/53 | 227/199 |
| Age | 83.7 (8.1) | 65.7 (13.7) |
| Diagnostic group | 52 NIA-Reagan Not-Low (MMSE 26.0 [4.3])<br>36 NIA-Reagan Intermediate-High (MMSE 20.6 [6.2]) | 264 Aβ- CU (MMSE 29.0 [1.2])<br>73 Aβ+ CU (MMSE 28.6 [1.4])<br>81 Aβ+ MCI (MMSE 26.5 [2.0])<br>8 Aβ+ AD dementia (MMSE 22.3 [2.7]) |
| APOE ε4 genotype | ε2/ε3 N = 8<br>ε2/ε4 N = 2<br>ε3/ε3 N = 49<br>ε3/ε4 N = 26<br>ε4/ε4 N = 3 | ε2/ε2 N = 2<br>ε2/ε3 N = 24<br>ε2/ε4 N = 10<br>ε3/ε3 N = 182<br>ε3/ε4 N = 180<br>ε4/ε4 N = 28 |

Continuous data are mean (standard deviation). Mini-Mental State Examination (MMSE) score is provided for the different diagnostic groups. NIA-Reagan refers to the National Institute on Aging-Reagan Institute criteria for the neuropathological diagnosis of Alzheimer disease (Hyman & Trojanowski, 1997). CU, cognitively unimpaired. MCI, mild cognitive impairment. AD, Alzheimer's disease.

P-tau217 as response variable and different sets of predictors, including I) only Aβ PET (using a large cortical region-of-interest [ROI]), II) only tau PET (using a global cortical ROI representing Braak stages I-VI), III) both Aβ PET and tau PET, and IV) Aβ PET and tau PET and their interaction. Model comparison of adjusted $R^2$ and AIC favored the model with the interaction term over the other models (Fig 5A). In the interaction model, both Aβ PET (β = 1.07, $P < 0.0001$) and tau PET (β = 0.44, $P < 0.0001$) and the Aβ by tau PET interaction (β = 0.31, $P < 0.0001$) were significantly associated with greater plasma P-tau217 (Aβ and tau PET were used as continuous variables, centered, and scaled to z-scores). Age ($P = 0.89$) and sex ($P = 0.91$) were non-significant predictors in the model. Higher Aβ PET was associated with higher plasma P-tau217 at all levels of tau PET (with strongest association in those with highest

tau PET) (Fig 5B). Higher tau PET was associated with higher plasma P-tau217 in those with high (but not in those with low) Aβ PET (Fig 5C) (but we acknowledge that the narrow range of tau PET in the individuals with low Aβ PET makes it more difficult to detect a significant association with plasma P-tau217, compared to in those with high Aβ PET, where the range of tau PET is wider). These results largely agreed with our results from the neuropathology dataset, showing that plasma P-tau217 is independently related to both Aβ and tau build-up.

## Plasma P-tau217 correlations with Aβ and tau PET stratified by Aβ and tau positivity

The finding that P-tau217 was independently related to both neuropathological densities of plaques and tangles, as well as both Aβ and tau PET measures, led us to the hypothesis that plasma P-tau217 mainly reflects Aβ pathology during the early stages of the disease (prior to significant neocortical spread of tangles), but is more closely related to tau pathology during the later stages of the disease (when both Aβ and tau pathologies are widespread). To test this hypothesis *in vivo*, we grouped the BioFINDER-2 participants into three groups: negative for both Aβ PET and tau PET ($N = 264$), Aβ PET-positive but tau PET-negative ($N = 104$), and positive for both Aβ and tau PET ($N = 58$). For this grouping, we used the tau PET uptake in a neocortical ROI corresponding to Braak regions I-IV (Leuzy *et al*, 2020). No individual in this dataset had negative Aβ PET combined with positive tau PET. Figure 5 (panels D-I) shows correlations between plasma P-tau217 and Aβ PET (Fig 5D–F) and tau PET (Fig 5G–I). There were no correlations between plasma P-tau217 and either Aβ or tau PET in the Aβ- and tau-negative group ($P > 0.05$, Fig 5D and 5G). In those with isolated Aβ positivity, plasma P-tau217 correlated with Aβ PET ($P < 0.001$, although the explanatory power of Aβ on plasma P-tau217 was modest, $R^2 = 0.10$, Fig 5E) but not tau PET ($P = 0.92$, $R^2 = 0$, Fig 5H). In the group positive for both Aβ and tau, plasma P-tau217 correlated with both Aβ PET ($R^2 = 0.14$ for Aβ, Fig 5F) and tau PET ($R^2 = 0.46$ for tau, Fig 5I) but had strongest correlation with tau PET. This supported our hypothesis that plasma P-tau217 reflects both Aβ and tau accumulation in AD, but primarily reflects Aβ accumulation during the early disease stages (when tau PET is still negative) and

**Figure 5. Associations between plasma P-tau217 with Aβ PET and tau PET.**

A $R^2$ and AIC for different regression models, using either only Aβ PET ("A"), only tau PET ("T(I-VI)"), both Aβ and tau PET ("A + T"), or both Aβ and tau PET including their interaction term ("AxT"). All models included age and sex as covariates. The panel shows adjusted $R^2$, together with AIC (above the bars) for each model. We compared $R^2$ between the models using a bootstrap procedure ($N = 1,000$ iterations), verifying that the $R^2$ for the "AxT" model was higher than for the "A + T" model ($\Delta R^2 = 0.026$, 95% CI 0.010-0.058), the "T(I-VI)" model ($\Delta R^2 = 0.17$, 95% CI 0.11-0.26), and the "A" model ($\Delta R^2 = 0.14$, 95% CI 0.060–0.22).

B Associations between plasma P-tau217 and Aβ PET stratified by tertiles (T) of tau PET (T1: SUVR ≤ 1.05, T2: 1.05 < SUVR ≤ 1.11, T3: 1.11 < SUVR ≤ 3.03).

C Associations between plasma P-tau217 and tau PET stratified by tertiles (T) of Aβ PET (T1: SUVR ≤ 0.468, T2: 0.468 < SUVR ≤ 0.579, T3: 0.579 < SUVR ≤ 1.08).

D–I Associations with P-tau217 across groups of Aβ and tau PET positivity. Cut-points for Aβ PET (> 0.533 SUVR) and tau PET (> 1.36 SUVR in a ROI corresponding to Braak stages I-IV) were used to define the groups. Associations are shown for global cortical Aβ PET (panels D-F) and global cortical tau PET (corresponding to Braak stages I-VI) (panels G-I) in individuals classified as negative for both Aβ and tau ($N = 264$, panels D and G), positive for Aβ only ($N = 104$, panels E and H), or positive for both Aβ and tau ($N = 58$, panels F and I). The plots show $R^2$ and $P$-values from linear regression models. No covariates were included in these models, in order to generate $R^2$-values for the PET measures alone.

Data information: In panels (B, C), tertiles were chosen to visualize the data, but the regression models used continuous PET uptake as predictors. In panel (C), one outlier was beyond the range of the x-axis (plasma P-tau217 = 17.8 ng/L, tau PET = 3.03 SUVR, 3rd Aβ PET tertile). The solid lines represent mean effects from regression models. The shaded areas are 95% confidence intervals for the mean effects. Samples were analyzed in duplicates.

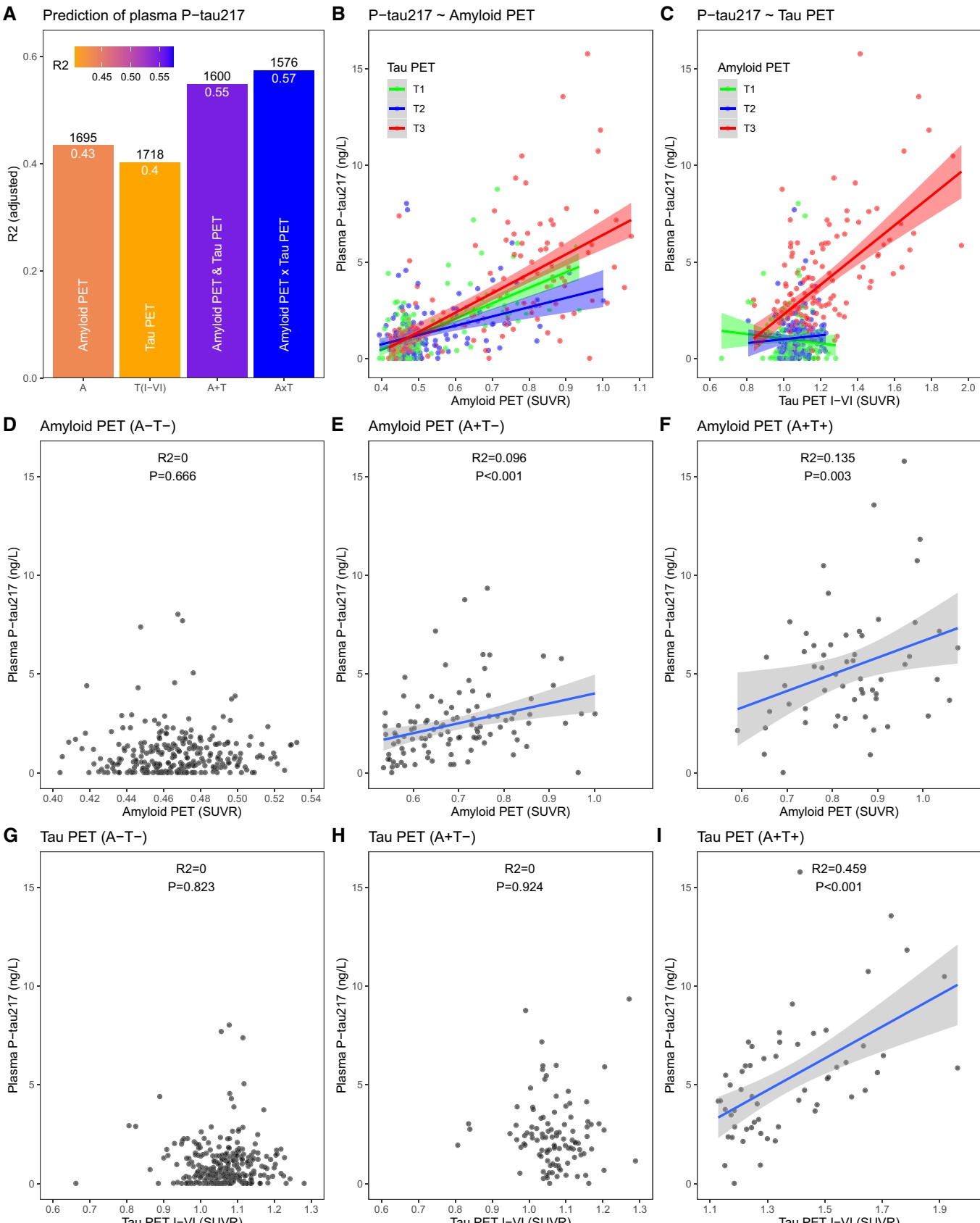

**Figure 5.**

primarily tau accumulation during the later disease stages (when both Aβ and tau PET are positive).

## Plasma P-tau217 statistically mediates effects of Aβ PET on tau PET

We also conducted statistical mediation analyses to test to what degree plasma P-tau217 explained effects of Aβ aggregation on tau uptake in BioFINDER-2. Plasma P-tau217 partly mediated the effect of Aβ PET on global tau PET (63% mediation, Fig 6A). The mediation was attenuated for tau PET in the medial temporal lobe (entorhinal cortex and hippocampus, 29%, Fig 6B). This is consistent with the hypothesis that the mechanism that links Aβ build-up to tau aggregation involves increased phosphorylation and/or release of soluble tau, and that this is especially important for tau aggregation that occurs outside of the medial temporal lobe (entorhinal cortex and hippocampus).

## Discussion

With recent advances in assay development, it is now possible to measure plasma levels of P-tau217 in clinical research, drug trials, and perhaps also in clinical practice in the future. To best interpret the P-tau217 results, it is important to understand what plasma P-tau217 levels represent. Although proposed frameworks have largely assumed that soluble P-tau best mirrors tau deposition (Jack et al, 2018), we and others have previously noted that CSF P-tau may be increased also as a function of Aβ deposition (Barthélemy et al, 2020b; Mattsson-Carlgren et al, 2020)—i.e., provide an indicator of Aβ-related tau pathophysiology that anticipates the development of measurable tau tangle pathology. We therefore hypothesized that plasma P-tau217 levels would be related to both Aβ and tau aggregation. Supporting our hypothesis, we found that both Aβ plaques and tau tangles were independently associated with higher plasma P-tau217 levels, when measured both post-mortem by neuropathological quantification (providing the most sensitive indicator of tau tangle deposition) and in vivo by PET imaging (which may not detect tau tangles until there is more substantial

and spatially extensive tangle deposition). Aβ plaques and tau tangles also interacted, so that highest plasma P-tau217 levels were seen in individuals who had high levels of both Aβ plaques and tau tangles. The independent effects of Aβ load and tau tangle load on plasma P-tau217 were significant, with considerable increases in model $R^2$ when combining Aβ and tau predictors compared to when using just one of them alone. Even though the results were mainly similar when using either post-mortem neuropathology examination or in vivo PET imaging to detect tau tangles, we found that plasma P-tau217 was more strongly associated with tau tangles than Aβ plaques when using neuropathology examination, which was not the case when using PET (compare Fig 1A with Fig 5A). This is probably explained by the fact that tau PET imaging does not reliably detect lower amounts of tau aggregates in the brain, as shown in a recent end-of-life study by Fleischer et al evaluating the diagnostic performance of 18F-flortaucipir (Fleisher et al, 2020). However, even when using neuropathology examination, Aβ pathology had an independent effect on plasma P-tau217 levels, and in cases with more limited tau pathology (mainly restricted to the medial temporal lobe), plasma P-tau217 correlated with Aβ pathology but not tau tangles, although we acknowledge that the methods used for tau quantification in the brain may not be sensitive to the earliest most subtle tau aggregation (Fig 3) (and since the total amount of aggregated tau is low in subjects with tau tangles limited to the medial temporal cortex, it is more difficult to detect an association between tangle load and plasma P-tau217 in those individuals compared to when tangles have also spread into neocortex). These results are congruent with recent studies showing that both blood and CSF P-tau217 are associated with Aβ PET also in tau PET-negative cases (Mattsson-Carlgren et al, 2020; Janelidze, 2021). Together, these results imply that plasma P-tau217 is not associated with tau tangle pathology independent from Aβ pathology, including primary age-related tauopathy (PART, see Fig 3B) (Sperling et al, 2014). However, plasma P-tau217 is associated with Aβ pathology, even during the early stages of the disease (Fig 3A), and it is further strongly associated with more widespread tau tangles in cases with Aβ pathology (Figs 1C and 4C). More research is needed to understand the molecular and cellular mechanisms behind how early Aβ pathology leads to changes in extracellular P-tau217 levels.

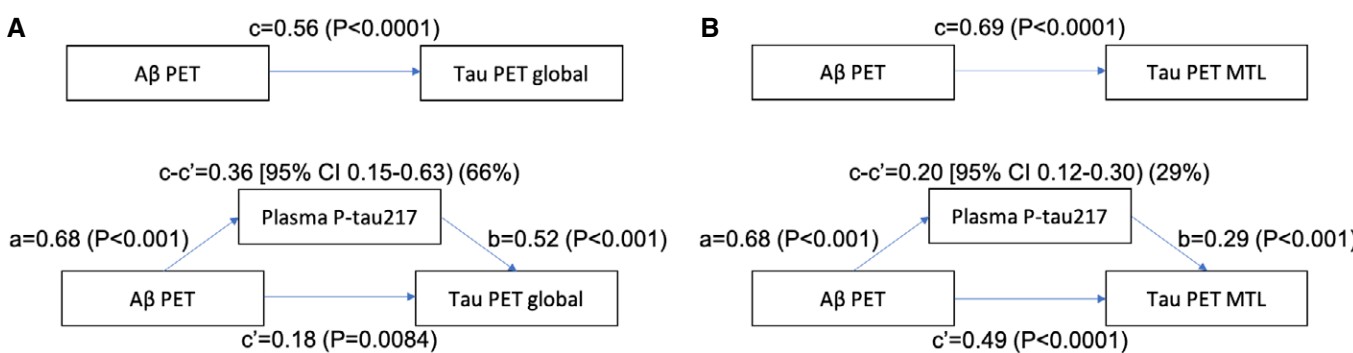

**Figure 6. Mediation analyses in BioFINDER-2.**

A, B  Plasma P-tau217 partly mediated the effect of Aβ PET on tau PET, when using tau PET quantified in Braak stages I-VI (A, "Tau PET global", 66% of the direct effect of Aβ PET on tau PET was explained by plasma P-tau217) or in the medial temporal lobe (B, MTL, entorhinal cortex and hippocampus, 29%). The P-values are from linear regression models used to estimate the mediation effects.

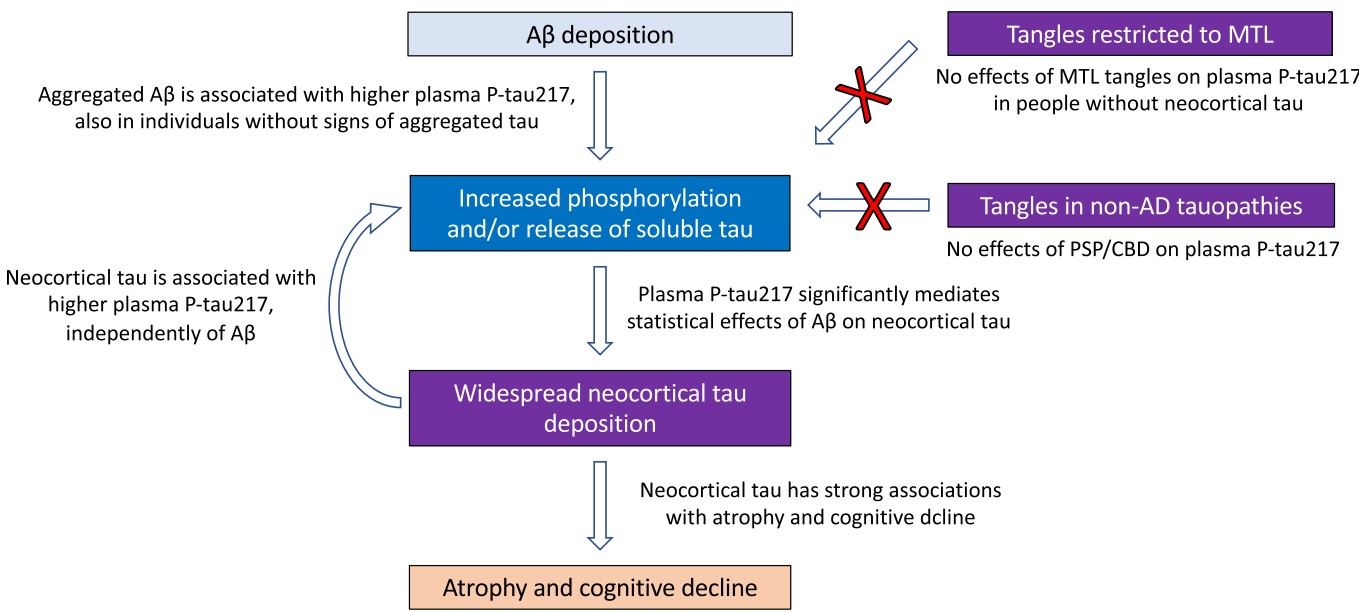

**Figure 7.  Hypothetical integration of neuropathology, fluid biomarker, and PET imaging findings.**

The figure shows hypothetical pathways, supported by our results, which link deposition of Aβ pathology to development of neocortical tangles through increased phosphorylation and/or release of soluble tau. We also include a hypothetical feedback loop, where tangle pathology also may drive increased levels of soluble P-tau, possible due to tau propagation between affected neurons. We also acknowledge that other alternative explanations may be considered for our findings. One possibility is that lack of sensitivity of our methods to accurately quantify aggregated tau makes us unable to detect very early associations between aggregated tau and plasma P-tau217.

Overall, the link between Aβ pathology and increased plasma P-tau217 is in line with recent results from cell models (Kwak *et al*, 2020) showing that phosphorylation of tau is increased in the presence of Aβ.

Plasma P-tau217 also provided statistical mediation for the effects of Aβ load on tau load in both the neuropathology dataset and the BioFINDER-2 PET imaging dataset. Intriguingly, the mediation effect of P-tau217 was especially strong for neuropathological quantification of tangle burden in the neocortex (excluding tangle count densities in entorhinal cortex and hippocampus, Fig 2C), where plasma P-tau217 explained up to 77% of the effect of Aβ plaques on tangle burden and the direct effect of Aβ plaques on tangles became non-significant when correcting for P-tau217. The in vivo measures with Aβ and global tau PET showed similar results, with considerable statistical mediation for plasma P-tau217 on global tau PET (66%), but less mediation (29%) when using tau PET quantified in the entorhinal cortex or hippocampus. This shows that plasma P-tau217 may be especially important for the spread of tau outside of the medial temporal lobe. Isolated medial temporal lobe tangle pathology (which may even appear in the absence of Aβ pathology, including "PART" (Crary *et al*, 2014)) appears not to be strongly related to increased phosphorylation and/or secretion of tau.

When grouping the subjects into three hypothetical "disease stages" (Fig 6), we found no correlations between plasma P-tau217 and either Aβ PET or tau PET in those that were negative for both Aβ and tau PET (we consider these individuals to *not* have AD), significant correlations to Aβ PET (but not tau PET) in those who were within the positive range for Aβ PET only (we consider this an early stage of AD), and significant correlations to both Aβ PET and tau PET (but strongest with tau PET) in those who were positive for both Aβ and tau PET (we consider this a later stage of AD). Taken together, these findings are congruent with the hypothesis that Aβ pathology leads to an increased release of soluble P-tau, which in turn is associated with a spread of tau tangle pathology beyond the medial temporal lobe. Figure 7 shows a hypothetical model which integrates the findings from this study and previous studies on biomarkers and development of AD. However, we note that different sensitivities of the biomarkers to detect underlying pathologies could impact their correlations. One possibility is that plasma P-tau217 may reflect AD-related tau hyperphosphorylation earlier and with higher sensitivity than tau PET. Alternatively, the plasma P-tau217 may initially increase due to increased site phosphorylation at 217 in the Aβ growth stage, and then as soluble total tau concentrations increase later, the P-tau217 increases further, but without specific increases in phosphorylation at site 217 (Barthélemy *et al*, 2020a, 2020b). Additional valuable information could come from Aβ overproducing mouse models that do not develop tau tangles, to characterize the extent to which Aβ could promote tau secretion even in the absence of subsequent tau tangle deposition (Maia *et al*, 2015; Mattsson-Carlgren *et al*, 2020).

Over the last years, there has been an increased interest in anti-tau treatments for AD (Bittar *et al*, 2020). One hypothetical possibility raised by our results is that therapies targeting the processes that lead to production of P-tau217 may break the link between build-up of aggregated Aβ and tau and thereby reduce atrophy and cognitive decline in AD (which are both strongly associated with tau pathology (Mattsson *et al*, 2019; Ossenkoppele *et al*, 2019)). At least one tau-directed treatment (using an anti-tau antibody) has been

reported to reduce CSF levels of P-tau217 (conference presentation by Michael Fresser at AAT-AD/PD 2020). Further drug development may incorporate plasma P-tau217 data in early stages to select drug candidates most likely to have beneficial effects on the biochemistry of tau in the brain in AD.

A limitation of this study is the relatively small dataset in the neuropathology cohort, with a very wide spread of different pathologies. Especially, there were quite few study participants with intermediate plaque counts (most had either very few plaques or substantial plaque pathology). Larger neuropathology datasets would be valuable to validate the findings directly in post-mortem material. We used the large BioFINDER-2 cohort to validate the findings, but we acknowledge that tau PET has limited sensitivity for early stage tau pathology (with isolated tangle pathology in entorhinal cortex and hippocampus). We therefore refrained from doing some of the subgroup analyses in BioFINDER-2 (focusing on MTL tau in individuals with negative global tau). Another limitation is that we used cross-sectional data, and longitudinal data for multiple modalities are needed to strengthen hypotheses about causal relationships between the biomarkers.

In summary, plasma P-tau217 is independently associated with both Aβ plaques and tau tangles, including associations with Aβ pathology even in cases with restricted tau tangle pathology and strong associations with tau tangles in Aβ-positive cases.

# Materials and Methods

### Participants

Study participants were from two cohorts. First, we included subjects from a biomarker-neuropathology cohort study (the Arizona Study of Aging and Neurodegenerative Disorders/Brain and Body Donation Program), also described previously (Palmqvist et al, 2020). Second, we included subjects from the prospective cohort study BioFINDER-2 (clinical trials: NCT03174938), explained previously (Palmqvist et al, 2020). From BioFINDER-2, we included all available cognitively unimpaired controls, Aβ+ mild cognitive impairment (MCI), and Aβ+ AD dementia patients (where Aβ+ was determined by Aβ PET, explained below) with Aβ and tau PET data. Study demographics are summarized in Table 1. The data used for this paper partly overlap with data in a previous publication in diagnostic performance of plasma P-tau217 (Palmqvist et al, 2020) (but the BioFINDER-2 sample used here is more restricted, since both Aβ and tau PET were required for inclusion in this study). Informed consent was obtained from all study participants.

### Fluid biomarkers

Plasma P-tau217 was measured using immunoassays at Lilly Research Laboratories, as described before (Palmqvist et al, 2020). In the Arizona Study of Aging and Neurodegenerative Disorders/ Brain and Body Donation Program, the mean (standard deviation) time between plasma collection and death was 1.2 (0.8) years. In BioFINDER-2, the mean time between plasma collection and PET imaging was 0.16 (0.22) years. Plasma P-tau217 data were not available to assessors of neuropathology or neuroimaging (minimizing subjective bias).

### Neuropathology

We included subjects from the biomarker-neuropathology cohort with or without neuropathological evidence of AD. Aβ plaque and neurofibrillary tangles were graded at standard sites in frontal, temporal, parietal, occipital cortices, the hippocampal CA1 region, and entorhinal/transentorhinal cortex, using 80-μm sections (Palmqvist et al, 2020). Plaque densities (including cored, neuritic, and diffuse plaques together) were derived using the Campbell–Switzer staining. Neurofibrillary tangle densities were derived using three staining methods (thioflavin S, Campbell–Switzer, and Gallyas) on the same sections. The CERAD templates (Mirra et al, 1991) were used to obtain semi-quantitative regional scores for Aβ and tangle pathology, which were summed to total plaque and tangle scores (range 0-15). All histopathological scoring was blinded to clinical or neuropathological diagnosis (see Table EV1 for details) and to biomarker levels.

### PET imaging

The procedures of tau PET (using RO948 labeled with radioactive fluorine [$^{18}$F]) and Aβ PET (using flutemetamol labeled with $^{18}$F) have been described previously (Palmqvist et al, 2020). Aβ PET was sampled in a global cortical ROI. When used as a dichotomous variable, Aβ PET > 0.533 SUVR was defined as a positive scan (defined previously using mixture modeling (Palmqvist et al, 2020)). Tau PET was sampled in different ROIs, corresponding to different Braak regions. For most analyses, we used a large composite corresponding to Braak regions I-VI, in order to comprehensively capture AD-related brain tau pathology. For some analysis, we used a dichotomous variable for neocortical tau PET positivity, using tau PET uptake in a more limited ROI (corresponding to Braak stages I-IV, where tau PET > 1.36 SUVR was defined as a positive scan (Leuzy et al, 2020)).

### Statistics

Associations between plasma P-tau217 and measures of Aβ and tau were tested in linear regression models, adjusted for age and sex. For models in the neuropathology cohort, we also evaluated models adjusted for the time between plasma sampling and death. This covariate was never significant and did not alter the effects of the other predictors and was therefore not included in the final models. Different regression models were compared on the same subjects and outcome, with different sets of predictors (including only Aβ data, only tau data, or both types of data, with or without an interaction term). Models were compared using $R^2$ and Akaike information criterion (AIC, where a lower value indicates a better fit, corrected for model complexity). Assumptions of regression models were assessed by inspection of standard diagnostic plots. Bootstrap procedures ($N = 1,000$ iterations) were used to compare $R^2$ between models. Mediation analyses were done to test the relationship between Aβ load, tau load, and plasma P-tau217 (using bootstrapped estimates of mediation effects). Statistics were done in R (v 4.0.2).

# Data availability

Anonymized data will be shared by request from a qualified academic investigator and as long as data transfer is in agreement

### The paper explained

#### Problem

Alzheimer's disease (AD) is characterized by the accumulation of β-amyloid and tau in the brain. Plasma levels of tau phosphorylated at threonine 217 (P-tau217) are increased in patients with AD, but it is not clear which pathophysiological process drives this biomarker change.

#### Results

Using a patient cohort with neuropathological quantification of brain pathologies and a cohort with *in vivo* positron emission tomography (PET) of β-amyloid and tau, we show that plasma P-tau217 had independent relationships with both β-amyloid and tau aggregation. Specifically, plasma P-tau217 appeared to be associated with β-amyloid also in individuals without signs of widespread neocortical tau pathology and mediated the association between aggregated β-amyloid and aggregated tau. In patients with both β-amyloid aggregation and neocortical tau pathology, plasma P-tau217 was strongly associated with neocortical tau. In patients with non-AD tauopathies (without β-amyloid pathology), there was no increase in plasma P-tau217.

#### Impact

Our study shows that accumulation of β-amyloid pathology may induce changes in phosphorylation and release of soluble tau, resulting in increased plasma P-tau217 before tau accumulation spreads in the neocortex. Later in the AD process, as tau aggregation spreads, plasma P-tau217 reflects the widespread accumulation of neocortical tau. Soluble phosphorylated tau may be investigated further to understand the pathobiological mechanisms of AD.

with EU legislation on the general data protection regulation and decisions by the Ethical Review Board of Sweden and Region Skåne, which should be regulated in a material transfer agreement.

**Expanded View** for this article is available online.

## Acknowledgements

Work at the authors' research center was supported by the Swedish Research Council (2016-00906), the Knut and Alice Wallenberg Foundation (2017-0383 and WCMM Fellowship for NMC 2019-2022), the Medical Faculty at Lund University and Region Skåne (WCMM Fellowship for NMC 2019-2022), the Marianne and Marcus Wallenberg Foundation (2015.0125), the Strategic Research Area MultiPark (Multidisciplinary Research in Parkinson's disease) at Lund University, the Swedish Alzheimer Foundation (AF-939932), the Swedish Brain Foundation (FO2019-0326, FO2020-0275), the Parkinson Foundation of Sweden (1280/20), the Skåne University Hospital Foundation (2020-O000028), Regionalt Forskningsstöd (2020-0314), the Konung Gustaf V:s och Drottning Victorias Frimurarestiftelse (2020 NMC), the Bundy Academy (Stora Priset 2020 NMC), and the Swedish federal government under the ALF agreement (2018-Projekt0279, 2018-Projekt0054). The Brain and Body Donation Program has been supported by the National Institute of Neurological Disorders and Stroke (U24 NS072026 National Brain and Tissue Resource for Parkinson's Disease and Related Disorders), the National Institute on Aging (P30 AG19610 Arizona Alzheimer's Disease Core Center), the Arizona Department of Health Services (contract 211002, Arizona Alzheimer's Research Center), the Arizona Biomedical Research Commission (contracts 4001, 0011, 05-901, 1001), the Arizona Department of Health Services (Grant No. CTR040636), and the Michael J. Fox Foundation for Parkinson's Research.

## Author contributions

Study design: NMC and OH. Data acquisition: SJ, RS, ES, SP, GES, TGB, EMR. Biochemical analyses: SJ, JLD. Neuroimaging analyses: RS. Funding acquisition: NMC, OH, EMR, TGB. Statistical analyses: NMC. Writing of first draft: NMC. Critical review of the manuscript: NMC, SJ, RJB, RS, ES, GES, EMR, SP, JLD, TGB, OH. Supervision: OH.

## Conflict of interest

OH has acquired research support (for the institution) from AVID Radiopharmaceuticals, Biogen, Eli Lilly, Eisai, GE Healthcare, Pfizer, and Roche. In the past 2 years, he has received consultancy/speaker fees from AC Immune, Alzpath, Biogen, Cerveau, and Roche. NMC, ES, SP, SJ, and RS have no disclosures. TGB has had research support from the National Institute on Aging, Michael J Fox Foundation for Parkinson's Research, and the State of Arizona and, in the past 2 years, has received consultancy and/or speaker fees from Prothena Biosciences and Vivid Genomics. JLD is an employee of Eli Lilly and Company. Remaining co-authors report no disclosures.

## For more information

See biofinder.se for more information about the study and cohort.

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
