## [Review Process File · EMBO Molecular Medicine]

Soluble P-tau²¹⁷ reflects amyloid and tau pathology and mediates the association of amyloid with tau

Niklas Mattsson-Carlgrén, Shorena Janelidze, RJ Bateman, Ruben Smith, Erik Stomrud, Geidy Serrano, Eric Reiman, Sebastian Palmqvist, Jeffrey Dage, Thomas Beach, and Oskar Hansson
DOI: [10.15252/emmm.202114022](https://doi.org/10.15252/emmm.202114022)

Corresponding authors: Niklas Mattsson-Carlgrén (niklas.mattsson-carlgrén@med.lu.se) , Oskar Hansson (oskar.hansson@med.lu.se)

Review Timeline:

Submission Date:	26th Jan 21
Editorial Decision:	16th Feb 21
Revision Received:	1st Mar 21
Editorial Decision:	16th Mar 21
Revision Received:	18th Mar 21
Accepted:	23rd Mar 21

Editor: Jingyi Hou

Transaction Report:

16th Feb 2021

Dear Dr. Mattsson,

Thank you again for submitting your work to EMBO Molecular Medicine. One of the referees who initially accepted to review the manuscript finally dropped out. We have now heard back from the other two referees who evaluated your manuscript. As you will see from the reports below, the referees acknowledge the potential interest of the study. However, they also raise a series of concerns about your work, which should be convincingly addressed in a major revision of the present manuscript.

The referees' recommendations are rather clear and there is no need to reiterate their comments. Importantly, alternative explanations need to be considered (as suggested by Referee #1) and overstatements should be avoided (commented by Referee #3). Referee #3 mentioned that additional data on pure tauopathy would strengthen the study, which we would encourage you to address.

We would welcome the submission of a revised version within three months for further consideration. Please note that EMBO Molecular Medicine strongly supports a single round of revision. As acceptance or rejection of the manuscript will depend on another round of review, your responses should be as complete as possible.

We are aware that many laboratories cannot function at full efficiency during the current COVID-19/SARS-CoV-2 pandemic and have therefore extended our "scooping protection policy" to cover the period required for a full revision to address the experimental issues. Please let me know should you need additional time, and also if you see a paper with related content published elsewhere.

I look forward to receiving your revised manuscript.

Sincerely,
Jingyi

Jingyi Hou
Editor
EMBO Molecular Medicine

*** Instructions to submit your revised manuscript ***

**** PLEASE NOTE **** As part of the EMBO Publications transparent editorial process initiative (see our Editorial at <https://www.embopress.org/doi/pdf/10.1002/emmm.201000094>), EMBO Molecular Medicine will publish online a Review Process File to accompany accepted manuscripts.

To submit your manuscript , please follow this link:

Link Not Available

When submitting your revised manuscript , please include:

- 1) a .docx formatted version of the manuscript text (including Figure legends and tables). Please make sure that the changes are highlighted to be clearly visible to referees and editors alike.
- 2) separate figure files*
- 3) supplemental information as Expanded View and/or Appendix. Please carefully check the authors guidelines for formatting Expanded view and Appendix figures and tables at <https://www.embopress.org/page/journal/17574684/authorguide#expandedview>
- 4) a letter INCLUDING the reviewers' reports and your detailed responses to their comments (as Word file)

Also, and to save some time should your paper be accepted, please read below for additional information regarding some features of our research articles:

- 5) The paper explained: EMBO Molecular Medicine articles are accompanied by a summary of the articles to emphasize the major findings in the paper and their medical implications for the non-specialist reader. Please provide a draft summary of your article highlighting
 - the medical issue you are addressing,
 - the results obtained and
 - their clinical impact.

- 6) For more information: There is space at the end of each article to list relevant web links for further consultation by our readers. Could you identify some relevant ones and provide such

information as well? Some examples are patient associations, relevant databases, OMIM/proteins/genes links, author's websites, etc...

7) Author contributions: the contribution of every author must be detailed in a separate section (before the acknowledgments).

8) EMBO Molecular Medicine now requires a complete author checklist (<https://www.embopress.org/page/journal/17574684/authorguide>) to be submitted with all revised manuscripts. Please use the checklist as a guideline for the sort of information we need WITHIN the manuscript as well as in the checklist. This is particularly important for animal reporting, antibody dilutions (missing) and exact p-values and n that should be indicated instead of a range.

9) Every published paper now includes a 'Synopsis' to further enhance discoverability. Synopses are displayed on the journal webpage and are freely accessible to all readers. They include a short stand first (maximum of 300 characters, including space) as well as 2-5 one sentence bullet points that summarise the paper. Please write the bullet points to summarise the key NEW findings. They should be designed to be complementary to the abstract - i.e. not repeat the same text. We encourage inclusion of key acronyms and quantitative information (maximum of 30 words / bullet point). Please use the passive voice. Please attach these in a separate file or send them by email, we will incorporate them accordingly.

You are also welcome to suggest a striking image or visual abstract to illustrate your article. If you do please provide a jpeg file 550 px-wide x 400-px high.

10) A Conflict of Interest statement should be provided in the main text

11) Please note that we now mandate that all corresponding authors list an ORCID digital identifier. This takes <90 seconds to complete. We encourage all authors to supply an ORCID identifier, which will be linked to their name for unambiguous name identification.

Currently, our records indicate that the ORCID for your account is 0000-0002-8885-7724.

Link Not Available

12) The system will prompt you to fill in your funding and payment information. This will allow Wiley to send you a quote for the article processing charge (APC) in case of acceptance. This quote takes into account any reduction or fee waivers that you may be eligible for. Authors do not need to pay any fees before their manuscript is accepted and transferred to our publisher.

Photos 400-800 DPI

*Additional important information regarding figures and illustrations can be found at <https://bit.ly/EMBOPressFigurePreparationGuideline>

***** Reviewer's comments *****

Referee #1 (Comments on Novelty/Model System for Author):

The dataset presented is exceptional, in particular the combination of neuropathological data with in vivo biomarker analysis in the same subjects. The addition of the in vivo PET data further increases the translational value of the study. Third, everything is nicely put together in a coherent model of AD pathogenesis.

Referee #1 (Remarks for Author):

Mattsson-Carlgren et al evaluated how 217phosphotau in plasma relates to neuropathological measures of tangles, plaques and their interaction and also to PET measures of these variables. They demonstrate that 217phosphotau depends on the amount of plaques, tangles and the interaction between the two. Medial temporal tangle density does not correlate with 217phosphotau levels while neocortical tangle density does. The results are interpreted within a model that is shown in Fig 7.

This is an exceptionally rich dataset with appropriate sample sizes. The report combines a set of postmortem cases of whom blood biomarker analysis pre-mortem is available with the well-known BioFinder cohort. The combination of neuropathology and in vivo blood biomarker analysis within the same subjects is particularly valuable. The data are impressive, in particular the neuropath+blood biomarker and the verification in vivo. My only concern relates to the interpretation in terms of the model outlined in figure 7. Other, more trivial explanations must be entertained.

Major comments:

1. Figure 3: The data are analysed with a linear regression but the distribution of the datapoints in panel A & B does not seem to be linear. The same is true for figure 1B for the pooled data. The authors should report whether the assumptions for linear regression are met.
2. The authors interpret their 217phosphotau data within a mainstream model that neocortical amyloidosis is a trigger for inducing the spread of medial temporal tangles to the neocortex. Here is an alternative explanation: Obviously the total amount of aggregated tau is much lower when tangles are limited to the medial temporal cortex than when tangles are also spreading into neocortex. Hence, if 217phosphotau reflects aggregated tau, it is more likely that a relationship will be found with neocortical than with medial temporal tau. Under this hypothesis the difference in the relationship of phosphotau to medial temporal versus neocortical tau is due to 1. a quantitative difference in total brain tangle load 2. the fact that neocortical spread of tau is rarely seen in amyloid negative individuals.
3. Likewise, on p 6 the interpretation of the observation that "Higher tau PET was associated with higher plasma P-tau217 in those with high (but not in those with low) A β PET", must take into account that those with a high Abeta PET have a much wider range in tau PET levels. When the

range of the predictor variable is wide, the likelihood to find a significant association if there is one is much higher than when the range of the predictor variable is narrow (as in the amyloid negative cases). The authors' interpretation that 'plasma P-tau217 is independently related to both A β and tau build-up' is interesting but an alternative, more trivial explanation should also be entertained.

Very minor comments:

1. Figure 3: when the linear regression does not reach significance there should be no regression line drawn in the plot

Referee #3 (Comments on Novelty/Model System for Author):

This is a well done study using state of the art biomarkers in humans it is sufficiently powered.

Referee #3 (Remarks for Author):

I think the data speaks for itself that pTau217 in blood is strongly associated with amyloid and tau pathology. This extends previous studies of this and other pTau epitopes.

However, association does not prove causality and the manuscript at times goes a little too far or is not precise enough with inference. One example is the statement:

"We also found that plasma P-tau217 levels mediated the effect of A β load on tau load."

This and a number of other similar interpretative statements should be modified to reflect that it is an association and as they did in the abstract say they potentially support a hypothesis.

A missing link here is data on a pure tauopathy (PSP, FTDP-17) it would strengthen the manuscript to include data on this group.

Overall I think a more conservative discussion is warranted but in general the study will be impactful on the field

Referee #1 (Comments on Novelty/Model System for Author):

The dataset presented is exceptional, in particular the combination of neuropathological data with in vivo biomarker analysis in the same subjects. The addition of the in vivo PET data further increases the translational value of the study. Third, everything is nicely put together in a coherent model of AD pathogenesis.

We are thankful to the reviewer for these positive comments.

Referee #1 (Remarks for Author):

Mattsson-Carlgrén et al evaluated how 217phosphotau in plasma relates to neuropathological measures of tangles, plaques and their interaction and also to PET measures of these variables. They demonstrate that 217phosphotau depends on the amount of plaques, tangles and the interaction between the two. Medial temporal tangle density does not correlate with 217phosphotau levels while neocortical tangle density does. The results are interpreted within a model that is shown in Fig 7.

This is an exceptionally rich dataset with appropriate sample sizes. The report combines a set of postmortem cases of whom blood biomarker analysis pre-mortem is available with the well-known BioFinder cohort. The combination of neuropathology and in vivo blood biomarker analysis within the same subjects is particularly valuable. The data are impressive, in particular the neuropath+blood biomarker and the verification in vivo. My only concern relates to the interpretation in terms of the model outlined in figure 7. Other, more trivial explanations must be entertained.

We agree, and we have added this text to the legend of figure 7: "We also acknowledge that other alternative explanations may be considered for our findings. One possibility is that lack of sensitivity of our methods to accurately quantify aggregated tau makes us unable to detect very early associations between aggregated tau and plasma P-tau217." We have also added other considerations throughout the text, as explained below.

Major comments:

1. Figure 3: The data are analysed with a linear regression but the distribution of the datapoints in panel A & B does not seem to be linear. The same is true for figure 1B for the pooled data. The authors should report whether the assumptions for linear regression are met.

We have inspected standard plots for diagnostic of the regression models. The linearity assumption was checked by inspection plots of model residuals versus fitted values. These did not indicate linearity problems. We include the plots below for the reviewer. We have also added text in the statistics section about inspection of diagnostic plots (page 15, line 11).

Residuals vs fitter values for model shown in Figure 1B:

Residuals vs fitter values for model shown in Figure 3A:

Residuals vs fitter values for model shown in Figure 3B:

2. The authors interpret their 217phosphotau data within a mainstream model that neocortical amyloidosis is a trigger for inducing the spread of medial temporal tangles to the neocortex. Here is an alternative explanation: Obviously the total amount of aggregated tau is much lower when tangles are limited to the medial temporal cortex than when tangles are also spreading into neocortex. Hence, if 217phosphotau reflects aggregated tau, it is more likely that a relationship will be found with neocortical than with medial temporal tau. Under this hypothesis the difference in the relationship of phosphotau to medial temporal versus

neocortical tau is due to 1. a quantitative difference in total brain tangle load 2. the fact that neocortical spread of tau is rarely seen in amyloid negative individuals.

We agree with this, and we have added a section in the discussion about this: (page 10, line 8) "(and since the total amount of aggregated tau is low in subjects with tau tangles limited to the medial temporal cortex, it is more difficult to detect an association between tangle load and plasma P-tau217 in those individuals compared to when tangles have also spread into neocortex)".

3. Likewise, on p 6 the interpretation of the observation that "Higher tau PET was associated with higher plasma P-tau217 in those with high (but not in those with low) A β PET", must take into account that those with a high Abeta PET have a much wider range in tau PET levels. When the range of the predictor variable is wide, the likelihood to find a significant association if there is one is much higher than when the range of the predictor variable is narrow (as in the amyloid negative cases). The authors' interpretation that 'plasma P-tau217 is independently related to both A β and tau build-up' is interesting but an alternative, more trivial explanation should also be entertained.

We agree, and we have added a section about this in the results (page 7, line 14): "(but we acknowledge that the narrow range of tau PET in the individuals with low A β PET makes it more difficult to detect a significant association with plasma P-tau217, compared to in those with high A β PET, where the range of tau PET is wider)".

Very minor comments:

1. Figure 3: when the linear regression does not reach significance there should be no regression line drawn in the plot

We have removed the regression lines from Figure 3B and Figure 3C (and from relevant panels in Figure 6).

Referee #3 (Comments on Novelty/Model System for Author):

This is a well done study using state of the art biomarkers in humans it is sufficiently powered.

We are grateful to the reviewer for this overall comment.

Referee #3 (Remarks for Author):

I think the data speaks for itself that pTau217 in blood is strongly associated with amyloid and tau pathology. This extends previous studies of this and other pTau epitopes.

However, association does not prove causality and the manuscript at times goes a little too far or is not precise enough with inference. One example is the statement: "We also found that plasma P-tau217 levels mediated the effect of A β load on tau load."

We have clarified that this was a statistical mediation, and changed it to “We also found that plasma P-tau217 levels statistically mediated the effect of A β load on tau load.” (page 3, line 18).

This and a number of other similar interpretative statements should be modified to reflect that it is an association and as they did in the abstract say they potentially support a hypothesis.

*We have made a number of changes throughout the text to modify statements that may be interpreted as too strong. We now point out in the introduction that this is an observational study (“the data from this observational study”, page 3, line 21). We mention at several places that the mediation effect represents “statistical mediation”. On page 5 (line 13) we have changed the sentence “These results **showed** that plasma P-tau217 may partly explain the link between build-up of A β and tau pathology” to “These results **suggest** that plasma P-tau217 may partly explain the link between build-up of A β and tau pathology”. We have also changed the sentence (page 5, line 25) “These results **show** that A β pathology is linked to increased phosphorylation and/or release of tau” to “These results **suggest** that A β pathology is linked to increased phosphorylation and/or release of tau”.*

A missing link here is data on a pure tauopathy (PSP, FTDP-17) it would strengthen the manuscript to include data on this group.

We have added data on 9 subjects with neuropathology data and pure tauopathies (PSP and CBD). These new data is described in the main text (page 6, line 6) and shown in the new Figure 4. These results showed that PSP/CBD patients (who did not have amyloid pathology) did not have increased plasma P-tau217 compared to individuals without significant primary pathologies. This supports that plasma P-tau217 is increased primarily in subjects with both amyloid and tau pathology. We have also added a sentence about this in the abstract: “P-tau217 was not elevated in patients with primary non-Alzheimer’s disease tauopathies (N=9).”

Overall I think a more conservative discussion is warranted but in general the study will be impactful on the field

We sincerely appreciate the positive comments. We have tried to make the discussion more conservative as described above.

16th Mar 2021

Dear Dr. Mattsson,

Thank you for the submission of your revised manuscript to EMBO Molecular Medicine. We have now received the enclosed report from the two referees who were asked to re-assess it. As you will see the referees are now supportive and I am pleased to inform you that we will be able to accept your manuscript pending the following amendments:

1. In the main manuscript file, please do the following:

- Provide up to 5 keywords and incorporate them in the main text
- Remove the red color font.
- Figure callouts: Please carefully check all callouts. Figure 4,a,b,c and Figure 5,d,e,f,g,h,i are not called out. Please fix these.
- Author contributions: Randall J. Bateman was not called out. The missing author should be included.
- The references need to be formatted according to the EMBO Molecular Medicine reference style. Please list up to 10 co-authors of a paper before adding et al. in the reference list/ citations should be listed in alphabetical order and list 10 co-authors of a paper before to add et al. Please remove weblinks.
- In Materials and Methods, include a statement that informed consent was obtained from all human subjects. Please make sure that the information entered in the Checklist regarding human subjects is also included in the Materials and Methods.

2. Funding information is incomplete in the online submission system -Please fix it.

3. Data Availability: Before submitting your revision, primary datasets produced in this study need to be deposited in an appropriate public database. The accession numbers and database should be listed in a formal "Data Availability "section (placed after Materials & Method) that follows the model below (see also <https://www.embopress.org/page/journal/17574684/authorguide#dataavailability>). Please note that the Data Availability Section is restricted to new primary data that are part of this study.

Data availability

4. Conflicts of interest: According to our editorial policy with regard to the "conflict of interest" (see below), the current statement suggests that you have no specific financial interest to declare - please confirm that.

'the journal requires authors of original research papers to declare any competing commercial interests in relation to the submitted work. It is difficult to specify a threshold at which a financial interest becomes significant, but as a practical guideline, we would suggest this to be any undeclared interest that could embarrass you were it to become publicly known.'

<https://www.embopress.org/page/journal/17574684/authorguide#conflictsofinterest>

5. Checklist:

- Both co-corresponding authors' names should be on the checklist.
- Please complete the 'E-human subjects', 'F-data accessibility' and 'G-dual use research of concern' parts.

6. For more information: There is space at the end of each article to list relevant web links for further consultation by our readers. Could you identify some relevant ones and provide such information as well? Some examples are patient associations, relevant databases, OMIM/proteins/genes links, author's websites, etc...

7. The Paper Explained: Please incorporate it into the main manuscript file. I have slightly modified the text. Please let me know if it is fine like this.

8. We would also encourage you to include the source data for figure panels that show essential data. Numerical data should be provided as individual .xls or .csv files (including a tab describing the data). For blots or microscopy, uncropped images should be submitted (using a zip archive if multiple images need to be supplied for one panel). Additional information on source data and instruction on how to label the files are available at

<https://www.embopress.org/page/journal/17574684/authorguide#sourcedata>

9. Our data editor has made a couple of suggestions about your manuscript (see attached). Please address these issues and keep the track mode on.

10. I have slightly modified and shortened the synopsis text. Please let me know if it is fine like this or if you would like to introduce future modifications.

Plasma levels of phosphorylated tau, including P-tau217, are elevated in Alzheimer's disease (AD). This study explores the underlying processes associated with the increased levels of plasma P-tau217, using post-mortem data and positron emission tomography (PET) of β -amyloid and tau.

- Plasma P-tau217 is independently associated with higher levels of both β -amyloid pathology and tau pathology in the brain.
- The first changes in plasma P-tau217 may reflect the early accumulation of β -amyloid before there is widespread tau aggregation.
- Once tau aggregation reaches the neocortex, there is a strong correlation between plasma P-tau217 and the amount of aggregated tau.
- Plasma P-tau217 mediates the association between β -amyloid accumulation and tau accumulation.

11. As part of the EMBO Publications transparent editorial process initiative (see our Editorial at <http://embomolmed.embopress.org/content/2/9/329>), EMBO Molecular Medicine will publish online a Review Process File (RPF) to accompany accepted manuscripts.

a. In the event of acceptance, this file will be published in conjunction with your paper and will

include the anonymous referee reports, your point-by-point response and all pertinent correspondence relating to the manuscript. Let us know if you do NOT agree with this.

I look forward to seeing a revised version of your manuscript as soon as possible.

Sincerely,
Jingyi

Jingyi Hou
Editor
EMBO Molecular Medicine

*** Instructions to submit your revised manuscript ***

To submit your manuscript, please follow this link:

Link Not Available

- 1) a .docx formatted version of the manuscript text (including Figure legends and tables)
- 2) Separate figure files*
- 3) supplemental information as Expanded View and/or Appendix. Please carefully check the authors guidelines for formatting Expanded view and Appendix figures and tables at <https://www.embopress.org/page/journal/17574684/authorguide#expandedview>
- 4) a letter INCLUDING the reviewer's reports and your detailed responses to their comments (as Word file).
- 5) The paper explained: EMBO Molecular Medicine articles are accompanied by a summary of the articles to emphasize the major findings in the paper and their medical implications for the non-

specialist reader. Please provide a draft summary of your article highlighting

6) For more information: There is space at the end of each article to list relevant web links for further consultation by our readers. Could you identify some relevant ones and provide such information as well? Some examples are patient associations, relevant databases, OMIM/proteins/genes links, author's websites, etc...

7) Author contributions: the contribution of every author must be detailed in a separate section.

8) EMBO Molecular Medicine now requires a complete author checklist (<https://www.embopress.org/page/journal/17574684/authorguide>) to be submitted with all revised manuscripts. Please use the checklist as guideline for the sort of information we need WITHIN the manuscript. The checklist should only be filled with page numbers where the information can be found. This is particularly important for animal reporting, antibody dilutions (missing) and exact values and n that should be indicated instead of a range.

9) Every published paper now includes a 'Synopsis' to further enhance discoverability. Synopses are displayed on the journal webpage and are freely accessible to all readers. They include a short stand first (maximum of 300 characters, including space) as well as 2-5 one sentence bullet points that summarise the paper. Please write the bullet points to summarise the key NEW findings. They should be designed to be complementary to the abstract - i.e. not repeat the same text. We encourage inclusion of key acronyms and quantitative information (maximum of 30 words / bullet point). Please use the passive voice. Please attach these in a separate file or send them by email, we will incorporate them accordingly.

You are also welcome to suggest a striking image or visual abstract to illustrate your article. If you do please provide a jpeg file 550 px-wide x 400-px high.

10) A Conflict of Interest statement should be provided in the main text

11) Please note that we now mandate that all corresponding authors list an ORCID digital identifier. This takes <90 seconds to complete. We encourage all authors to supply an ORCID identifier, which will be linked to their name for unambiguous name identification.

Currently, our records indicate that the ORCID for your account is 0000-0002-8885-7724.

Link Not Available

12) The system will prompt you to fill in your funding and payment information. This will allow Wiley to send you a quote for the article processing charge (APC) in case of acceptance. This quote takes into account any reduction or fee waivers that you may be eligible for. Authors do not need to pay any fees before their manuscript is accepted and transferred to our publisher.

Photos 400-800 DPI

*Additional important information regarding figures and illustrations can be found at <https://bit.ly/EMBOPressFigurePreparationGuideline>

The system will prompt you to fill in your funding and payment information. This will allow Wiley to send you a quote for the article processing charge (APC) in case of acceptance. This quote takes into account any reduction or fee waivers that you may be eligible for. Authors do not need to pay any fees before their manuscript is accepted and transferred to our publisher.

***** Reviewer's comments *****

Referee #1 (Comments on Novelty/Model System for Author):

As described in my previous review

Referee #1 (Remarks for Author):

The authors addressed my comments satisfactorily.

Referee #3 (Comments on Novelty/Model System for Author):

its analysis and humans

Referee #3 (Remarks for Author):

I think the manuscript which was sound to begin with is now improved by being more precise with the semantics. The title reminds somewhat misleading though perhaps consider Soluble P-tau217 reflects amyloid and tau pathology and reflects the association of amyloid with tau

An important study!

The authors performed the requested editorial changes.

23rd Mar 2021

Dear Dr. Mattsson-Carlgrén,

We are pleased to inform you that your manuscript is accepted for publication and is now being sent to our publisher to be included in the next available issue of EMBO Molecular Medicine.

We would like to remind you that as part of the EMBO Publications transparent editorial process initiative, EMBO Molecular Medicine will publish a Review Process File online to accompany accepted manuscripts. If you do NOT want the file to be published or would like to exclude figures, please immediately inform the editorial office via e-mail.

Please read below for additional IMPORTANT information regarding your article, its publication and the production process.

Congratulations on your interesting work,

Jingyi

Jingyi Hou
Editor
EMBO Molecular Medicine

Follow us on Twitter @EmboMolMed
Sign up for eTOCs at embopress.org/alertsfeeds

Corresponding Author Name: Niklas Mattsson-Carlgrén and Oskar Hansson

Manuscript Number: EMM-2021-14022